# Cation desolvation-induced capacitance enhancement in reduced graphene oxide (rGO)

Kangkang Ge[1], Hui Shao[2], Encarnacion Raymundo-Piñero[3,4], Pierre-Louis Taberna [1,3] ✉ & Patrice Simon [1,3] ✉

Understanding the local electrochemical processes is of key importance for efficient energy storage applications, including electrochemical double layer capacitors. In this work, we studied the charge storage mechanism of a model material - reduced graphene oxide (rGO) - in aqueous electrolyte using the combination of cavity micro-electrode, operando electrochemical quartz crystal microbalance (EQCM) and operando electrochemical dilatometry (ECD) tools. We evidence two regions with different charge storage mechanisms, depending on the cation-carbon interaction. Notably, under high cathodic polarization (region II), we report an important capacitance increase in $Zn^{2+}$ containing electrolyte with minimum volume expansion, which is associated with $Zn^{2+}$ desolvation resulting from strong electrostatic $Zn^{2+}$-rGO interactions. These results highlight the significant role of ion-electrode interaction strength and cation desolvation in modulating the charging mechanisms, offering potential pathways for optimized capacitive energy storage. As a broader perspective, understanding confined electrochemical systems and the coupling between chemical, electrochemical and transport processes in confinement may open tremendous opportunities for energy, catalysis or water treatment applications in the future.

Understanding the charge storage mechanism and, more generally, the processes occurring at the electrode/electrolyte interface, is of key importance for the development of electrochemical energy storage material and devices with improved performance. Unlike batteries, where diffusion limitations in the electrodes are prevalent, charge storage in electrochemical double layer capacitors is governed by a surface-controlled process, thus offering high-rate charge-discharge capability. Historically, the electric double layer (EDL) classical model proposed by Helmholtz and further refined by Gouy–Chapman–Stern (GCS) in the 19th century set the scientific foundations for understanding the charge/discharge mechanisms in capacitive materials[1]. Classical EDL-type charge storage mechanism is achieved by ion adsorption/desorption through electrostatic forces without involving any redox charge transfer. This characteristic ensures fast charging rate, no diffusion limitation and long-term cycling stability[2]. Pseudocapacitive electrodes exhibit similar electrochemical response to EDL-type electrodes like porous carbon and graphene, with a charge changing linearly with the potentials[3,4]. However, pseudocapacitive charge storage involves charge transfer or partial charge transfer across the electrochemical interface, distinguishing such process from the electrostatic capacitive charge storage observed in EDL-type electrodes[4], which in turn leads to a lower cycling stability. Both capacitive and pseudocapacitive electrodes exhibit unique confined geometries, such as narrow slit pores for porous carbon or layered

[1]Université Paul Sabatier, CIRIMAT UMR CNRS 5085, 118 Route de Narbonne, 31062 Toulouse, France. [2]i-Lab, CAS Center for Excellence in Nanoscience, Suzhou Institute of Nano-Tech and Nano-Bionics (SINANO), Chinese Academy of Sciences (CAS), Suzhou 215123, China. [3]Réseau sur le Stockage Electrochimique de l'Energie (RS2E), FR CNRS 3459 Amiens, France. [4]Université Orléans, CNRS, CEMHTI UPR3079 Orléans, France. ✉e-mail: pierre-louis.taberna@univ-tlse3.fr; patrice.simon@univ-tlse3.fr

structures for MXenes and graphene-like materials. Confined geometry arises when the traditional planar interfaces approach to each other, leading to an overlap of EDLs[5]. Over the past several decades, significant advancements have been made regarding the characterization of nanoporous carbons for capacitive energy storage[6,7]. A notable finding was the anomalous increase in capacitance when pore sizes were reduced to below 1 nm[8]. Smaller pore radii permit ions to get closer to the carbon pore walls thanks to partial desolvation, resulting in the creation of image charges in the carbon electrodes, more efficient screening the ionic charges and capacitance increase[9,10]. As a result, enhanced energy storage capabilities are achieved, suggesting that energy stored at the interface is somehow directly correlated to solvation energy[11,12].

Confined geometries such as narrow slit pores are also present in 2D layered materials, including reduced graphene oxide (rGO), MXenes, and others[13]. In particular, water confined between the interlayers of 2D materials is often reported to show improved charge transport kinetics and efficient charge storage in layered metal oxides[14,15], including $MnO_2$[16,17], $V_2O_5$[18], $WO_3$[19,20] and so on. Specifically, Augustyn et al. emphasized the impact of confined structural water networks in $WO_3 \cdot nH_2O$, which promotes the structural stability and facilitates a rapid proton intercalation rate[19]. Additionally, the promotional effect of structural water has been reported in realizing significant capacity improvements in vanadium oxide nanosheets[18] and $MnO_2$ birnessite nanolayers, where the intercalation of partial desolvated $K^+$ ions resulted in improved capacity by increasing charge transfer to the host[14]. These recent studies suggest that capacitive charge storage under confined geometries cannot be simply described as either pure EDL or pseudocapacitance, but instead should be considered more as a continuum based on the solvent-mediated interactions between electrolyte ions and electrode host materials[13,17].

For 2D layered materials, the nature of surface terminations has also been identified as a key factor in affecting charge storage capability. For instance, oxygen-containing surface functional groups significantly improve charge storage through efficient surface redox reactions in aqueous electrolytes[21,22]. Additionally, interlayer confined water is controlled by the presence of –OH functional groups on MXenes, which plays a pivotal role in facilitating the transport of hydrated $H^+$ in the interlayer spacing to reach the redox sites[23]. Besides, efficient ion transport was observed with nitrogen-terminated MXenes, still in acidic aqueous electrolytes[24]. However, most of the studies deal with acidic/alkaline electrolytes, basically because of their high ($H^+$ and $OH^-$) ionic conductivity and chemical reactivity, leaving vast array of neutral electrolytes using metallic salts largely unexplored. One of the challenges is that electrolyte ions cannot simply be treated as point charges. For instance, the adsorption of cations with different valence and solvated ion sizes including $Na^+$, $Zn^{2+}$ and $Al^{3+}$, has been studied in highly ordered and compact porous electrodes[25]. The partially dehydrated divalent $Zn^{2+}$ ion was found to be densely and neatly packed, achieving an unparalleled spatial charge density by balancing the valence and size of charge-carrier ions. Drawing parallels with the findings in porous electrodes, one might wonder whether the ion desolvation is the key for optimizing energy storage in 2D layered conducting materials. To understand the charge storage at confined nanoscale interface, a systematic exploration and deeper comprehension are highly desired.

The present paper provides evidence of the key role of ion partial desolvation on the charge storage mechanism in 2D rGO taken as model material. Thanks to an original combination of several operando techniques including electrochemical quartz crystal microbalance (EQCM) and electrochemical dilatometry (ECD) measurements, we evidenced partial desolvation of cations during cathodic polarization, resulting in enhanced capacitance. In this paper, multilayer rGO particles with conducting graphitic domains and controllable surface oxygen functionalities were used as model materials

to study charge storage mechanism in near-neutral aqueous electrolytes and the effect of ion confinement on the electrochemical behavior. Preliminary studies have already shown a cation-dominated charging mechanism for largely reduced graphene oxide materials[26]. Ion adsorption kinetics and electrochemical behaviors were also found to be different when changing the type of cation used[25–27]. Key parameters such as charge density, hydration/solvation energies, and specific cation-graphene interactions - like the cation-$\pi$ interaction - influence interfacial electrochemical activity, making the capacitive charge storage mechanisms of (2D layered) rGO materials more intricate than anticipated.

Here, we aim at clarifying how the ion-carbon interactions and ion desolvation affect the electrochemical behavior of rGO. The electrochemical behavior of rGO electrode was studied in three different electrolytes containing cations including $Li^+$, $Mg^{2+}$, and $Zn^{2+}$. Cyclic voltammetry experiments were firstly achieved by using a cavity micro-electrode (CME) setup, which allows for tracking tiny changes in the electrochemical signature, and improves the signal/noise ratio and mitigates ohmic drop. Then, operando EQCM experiments have been used to bring a better understanding about ion fluxes and ion solvation-desolvation occur during the adsorption-desorption process at the rGO surface. Eventually, the operando ECD experiments were made and cross-correlated with EQCM data to better grasp the charge adsorption mechanism in rGO.

## Results
### Synthesis and physicochemical characterization of rGO
The preparation of the materials is illustrated in Fig. 1a. rGO was synthesized from a graphene oxide (GO) aqueous dispersion at a concentration of 1 mg mL$^{-1}$ using a mild hydrothermal reaction at 160 °C for 6 h[28]. Subsequently, to eliminate the residual surface oxygen, a second thermal reduction of the rGO powder was conducted at 900 °C under an Ar atmosphere for 1 h. This product, obtained after a double reduction of GO, is denoted as r$^2$GO. Figure 1b displays the X-ray diffraction (XRD) patterns of both rGO and r$^2$GO powders. The successful synthesis of rGO was validated by the presence of distinct XRD peaks at $2\theta = 24.0°$, corresponding to a d-spacing along the c-axis of 3.7 Å for the (002) diffraction plane and $2\theta = 43.8°$ for the (100) plane. The d-spacing of the (002) plane reflects the interlayer distance between individual graphene layers. The marginally enlarged interlayer distance in our sample may result from the heightened presence of surface functional groups. For r$^2$GO, the interlayer distance decreases to 3.4 Å at $2\theta = 26.2°$, which is attributed to the elimination of various functional groups originally situated between the rGO interlayers[29]. To identify the oxygen content of rGO, Supplementary Fig. 1 shows the mass loss of rGO during the two-step thermal reduction processes by temperature programmed desorption mass spectrometry (TPD-MS). TPD-MS results indicate that considerable amount of oxygen surface functionalities still exist on the rGO surface after hydrothermal reduction of GO[30]. Indeed, after quantification of the spilled gas product such as CO and $CO_2$, the content of surface oxygen is determined as 16.7 wt.% for rGO and 1.3 wt.% for the further reduced sample r$^2$GO.

Due to the residual oxygen-containing surface groups, rGO consistently retains a negative charge in aqueous solutions when the solution's pH exceeds 3[31]. This is confirmed by the zeta potential values of rGO samples in various electrolytes (Fig. 1c), which are negative whatever the electrolyte that has been used. More precisely, we changed the cation type to explore the cation-rGO interactions, given earlier indications that the charge storage mechanism of rGO is predominantly cation-controlled[26,27]. We maintained the anion as $Cl^-$ with electrolyte pH≈6 in 0.01 M concentration for zeta potential measurements. The zeta potential for rGO in the presence of LiCl electrolyte was $-37 \pm 4$ mV, consistent with already reported rGO zeta measurement at different pH[31]. Nevertheless, the reported results are focused on the influence of pH without paying attention to the role of cation.

According to our zeta measurements, it turns out that electrolytes containing divalent cations drastically altered the surface charges, as zeta potentials of $-11 \pm 1$ mV and $-5 \pm 1$ mV were measured in $MgCl_2$ and $ZnCl_2$ electrolytes, respectively. The important noncovalent cation-π interaction between cations and the delocalized aromatic graphene domains distributed in rGO[32] are supposed to be at the origin of the change in the surface charges of rGO[33]. In the case of $Mg^{2+}$, the higher charge density as a divalent cation[34,35] explains the higher adsorption energy induced by cation-π interaction versus $Li^+$[35,36]. Therefore, the enrichment of $Mg^{2+}$ onto the rGO surface is expected to effectively screen the surface charge and makes the surface zeta potential less negative than $Li^+$. Interestingly, a stronger metal ion-π interaction[37–39] were observed with transition metal ions - $Zn^{2+}$ in our study - resulting in further decrease of the rGO surface charge. Thus, according to those zeta measurements, the order of surface adsorption at open circuit potential is: $Zn^{2+} > Mg^{2+} > Li^+$.

## Cation-dependent electrochemical signature of rGO

In light of the rapid charging kinetics associated with capacitive materials, traditional studies often overlook the tiny differences among electrolytes with similar physiochemical properties; however, we try to address this point in the present work. Figure 2a illustrates a three-electrode setup where a CME is used as the working electrode. The CME consists in a platinum wire encased in a silica glass tube. A small amount of active powder material ($10^{-7}$–$10^{-8}$ g) is compacted into a cavity of approximately 50 μm in diameter and 20 μm in depth, eliminating the need for conductive agents and binders. Thanks to such electrode design, the measured current is less than few μA, significantly mitigating ohmic drop, which allows for obtaining cyclic voltammograms with better definition. Therefore, we take benefit of this set-up to perform a more accurate electrochemical signature analysis of rGO in three different electrolytes: LiCl, $MgCl_2$, and $ZnCl_2$ (Fig. 2b–d).

As observed in Fig. 2b, in the $Li^+$-based electrolyte, there are two current regions: one located between +0.7 V down to +0.4 V vs. Ag/AgCl, with the other between +0.4 V and −0.8 V vs. Ag/AgCl. In both regions, the current plateaus but at different magnitude. According to

some other studies, such current change giving rise to a higher current amplitude at higher overpotentials is highly likely correlated with cation desolvation[27,40,41]. Interestingly, comparing the divalent $Mg^{2+}$-based electrolyte to the $Li^+$-based electrolyte (Fig. 2c), additional pair of current bumps located at ~ −0.5 V vs. Ag/AgCl is seen in the CV. Meanwhile, a broad current bump emerges in the potential window of −0.5 V to +0.2 V vs. Ag/AgCl for the $Zn^{2+}$-based electrolyte (Fig. 2d)[42]. According to those results, as in each electrolyte anions are the same, these patterns are highly likely correlated with the cation type; moreover, it has also been found out the type of anion does not affect significantly the electrochemical signature (Supplementary Fig. 2). We supposed that the interactions between cations and the rGO seems then to be crucial in controlling the (cation-dominated) electrochemical responses. We further explored the charging kinetics by plotting the logarithm of the bump/peak current against the scan rate. The results suggest a capacitive charging mechanism as current linear increase with scan rate[43] (Supplementary Fig. 3). It is worth mentioning, after surface oxygen functionalities have been removed ($r^2GO$ sample), the capacitive current was divided by 2 to 5 (Supplementary Fig. 4), which is attributed to the decrease of surface wettability, and all the CVs display a rectangular shape without any current peaks or bumps. Importantly, the pH of the electrolytes (1 M) shown in Fig. 2 is 6.9 for LiCl, 5.0 for $MgCl_2$, and 5.3 for $ZnCl_2$. These values are near neutral, despite the slight hydrolysis of multivalent metal cations in water. As a result, cations seem to act as the main charge carrier in our case instead of the hydrated protons, differentiating our study from those examining the faradaic reversible reactions between protons and carbon surface oxygen-containing functionalities.

To further validate the cation- and oxygen-dependent electrochemical behavior, we conducted supplementary Swagelok cell experiments with mass loadings of about 0.7 mg cm$^{-2}$. CVs shown in Supplementary Fig. 3 mirror a similar trend: while current bumps are present in rGO, they disappear for $r^2GO$. Furthermore, we calculated the capacitance of both rGO and $r^2GO$ (Supplementary Table 1) across the different electrolytes by integrating the CV with Eq. (1). As expected from previous works[44], we observed that surface oxygen functionalities considerably enhance capacitance, most possibly attributed

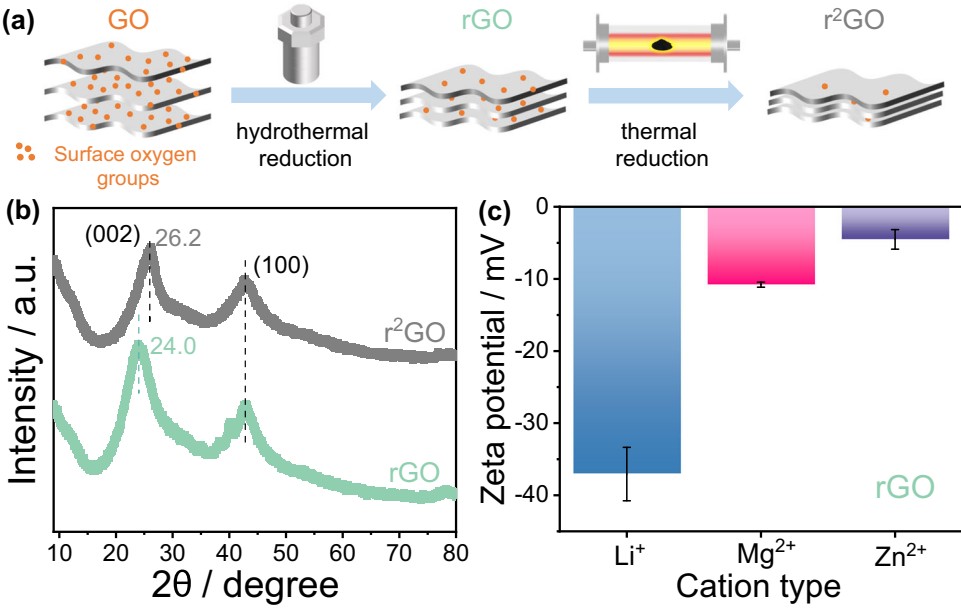

**Fig. 1 | Material preparation and physiochemical properties characterization.** **a** Experimental procedures of the step reduction of graphene oxide (GO): reduced graphene oxide (rGO) was obtained by first hydrothermal reduction and followed by high-temperature thermal reduction to prepare further reduced material (marked as $r^2GO$). **b** X-ray diffraction (XRD) patterns of rGO and $r^2GO$ powder. The 2θ angle corresponding to the 002 crystal plane XRD peaks of GO is shifted from 24.0° (d-spacing 3.7 Å) to 26.2° (d-spacing 3.4 Å) after reduction. **c** Zeta potential of rGO in the presence of aqueous 0.01 M LiCl, $MgCl_2$ and $ZnCl_2$ solutions.

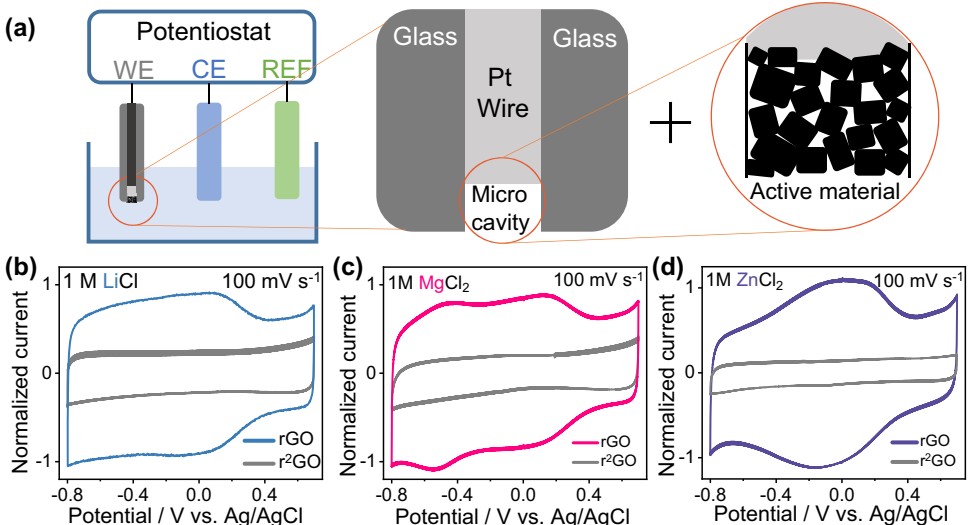

**Fig. 2 | Electrochemical response of rGO in three different electrolytes.** Experimental set-up and electrochemical behavior of rGO in different neutral aqueous electrolytes in a three-electrode configuration with a cavity microelectrode (CME) serving as working electrode. **a** illustration of the three-electrode set-up: the active material (rGO or r²GO) is encapsulated in the cavity (50 μm diameter and 20 μm depth). Cyclic voltammetry curves of rGO and r²GO in the electrolyte of (**b**) 1 M LiCl, (**c**) 1 M MgCl₂, and (**d**) 1 M ZnCl₂ at a scan rate of 100 mV s⁻¹.

to increased exposure of active sites and enhanced ion transport due to improved wetting capabilities.

## Two regions of cation solvation-desolvation

EQCM stands as an effective operando technique for monitoring ionic fluxes at polarized electrode interfaces. According to the Sauerbrey equation[45], the electrode mass change ($\Delta m$) arising from the electrochemical ion adsorption/desorption in/from the carbon electrode is negatively correlated to the change of the quartz resonant frequency ($\Delta f$) (see Eq. (2)): a decrease in $\Delta f$ indicates an increase in $\Delta m$ and vice versa. Figure 3a–c show the CV curves in three electrolytes obtained using the EQCM cell, which are similar to those previously obtained with CMEs. As a whole, in all plots, the frequency change $\Delta f$ decreases ($\Delta m$ increases) during cathodic polarization and $\Delta f$ increases ($\Delta m$ decreases) during anodic polarization. Globally speaking, the overall trend of $\Delta f$ keeps the same regardless of cation type.

The next part is focused on the detailed analysis of the $\Delta f$ curves in conjunction with both the potential and the cation need to be further analyzed. The cathodic polarization span, ranging from 0.5 V to −0.8 V vs. Ag/AgCl, was chosen for a deeper examination of the cation-dependent charge storage mechanism. According to the minimal change in motional resistance ($\Delta R$) shown in Supplementary Fig. 5, the coating onto the quartz can be considered as rigid so that the gravimetric model of the Sauerbrey equation remains valid. $\Delta m$ was plotted against the accumulated charges ($\Delta Q$) as depicted in Fig. 3d–f. $\Delta Q$ was calculated from the integration of the CV curves (current over time) and was normalized to zero at 0.5 V vs. Ag/AgCl, where the potential of zero charge seems to be. $\Delta m$ increases with the accumulated cathodic charges, basically suggesting a cation adsorption mechanism, which is in a good agreement with other EQCM studies that reported a predominantly cation-driven charging mechanism for alkali metal ions and rGO[26,27]. Two distinct regions can be defined from the $\Delta m$-$\Delta Q$ plots (Fig. 3d–f): Region I displays a steady growth with a constant slope, while Region II shows a smooth transition and a reduced slope. These regions are particularly pronounced for divalent $Zn^{2+}$. Generally, the steady mass increase at low polarized potentials in Region I, driven by cation adsorption, is shared by the three cations, but with different slopes. Interestingly, the charge storage in Region II is more efficient considering less mass is required to balance equivalent $\Delta Q$ at large polarizations when compared to Region I. Regarding the cations, especially multivalent ones, they are known to be highly hydrated in bulk electrolyte due to their high charge density and solvation free energy. So, a possible explanation of the slope decrease of the $\Delta m$-$\Delta Q$ plots at high charge (large polarizations) points towards cation dehydration.

To have a deeper look into the ion dehydration process during cathodic polarization (cation adsorption) and clarify the relationship between ion dehydration and the cation-dependent current responses, we used multi-linear fitting of the $\Delta m$-$\Delta Q$ plots. Each calculated slope within the linear fitting scale is interpreted as a molecular weight according to Faraday's law (Eq. (3)), as illustrated in Fig. 3g–i. The white dashed lines show the multi-linear fitting results, with the cut-off potentials for each scale indicated by vertical black dashed lines. Finally, we plotted both the current and the calculated molecular weight versus the electrode potential to establish the correlation between the observed current bumps and ion dehydration.

In Region I, all three cations exhibited a similar trend: an increasing current paralleled by cation adsorption with a stable molecular weight. The molecular weight of these hydrated cations reveals varying hydration levels among the cations. For instance, the average solvation number for monovalent $Li^+$ ion is 4.6 (Fig. 3g), while $Mg^{2+}$ (Fig. 3h) and $Zn^{2+}$ (Fig. 3i) average 18.4 and 12.7 $H_2O$ molecules, respectively. Interestingly, the idea of each $Li^+$ carrying 4–5 $H_2O$ molecules at mild polarization was also proposed from electrogravimetric impedance studies[27]. Due to higher hydration free energy for multivalent cations, the average solvation number for $Mg^{2+}$ is found to be as high as 18.4, which well aligns with previous reports with a value of 18, counting both the primary coordination number (6) and the secondary solvation shell number (12)[46,47]. In the case of $ZnCl_2$ electrolyte, 75% of $Zn(H_2O)_n^{2+}$ likely serves as the primary charge carrier (25% of $Zn(H_2O)_nCl^+$) in 1 M $ZnCl_2$ solutions, while the potential formation of zinc clusters usually occurs in highly concentrated $ZnCl_2$ solutions[48,49]. Besides, the electrochemical response shows no anion dependence when altering the anion including $NO_3^-$ and $SO_4^{2-}$ (Supplementary Fig. 2), which also evidences that the electrochemical response are related to cation $Zn^{2+}$ but not the zinc clusters. Given the comparable hydration free energy of $Zn^{2+}$ (−1995 kJ mol⁻¹) to $Mg^{2+}$ (−1830 kJ mol⁻¹)[25], an average solvation number of 12.7 possibly include the primary hydration shell (6) and extended hydration shells[50].

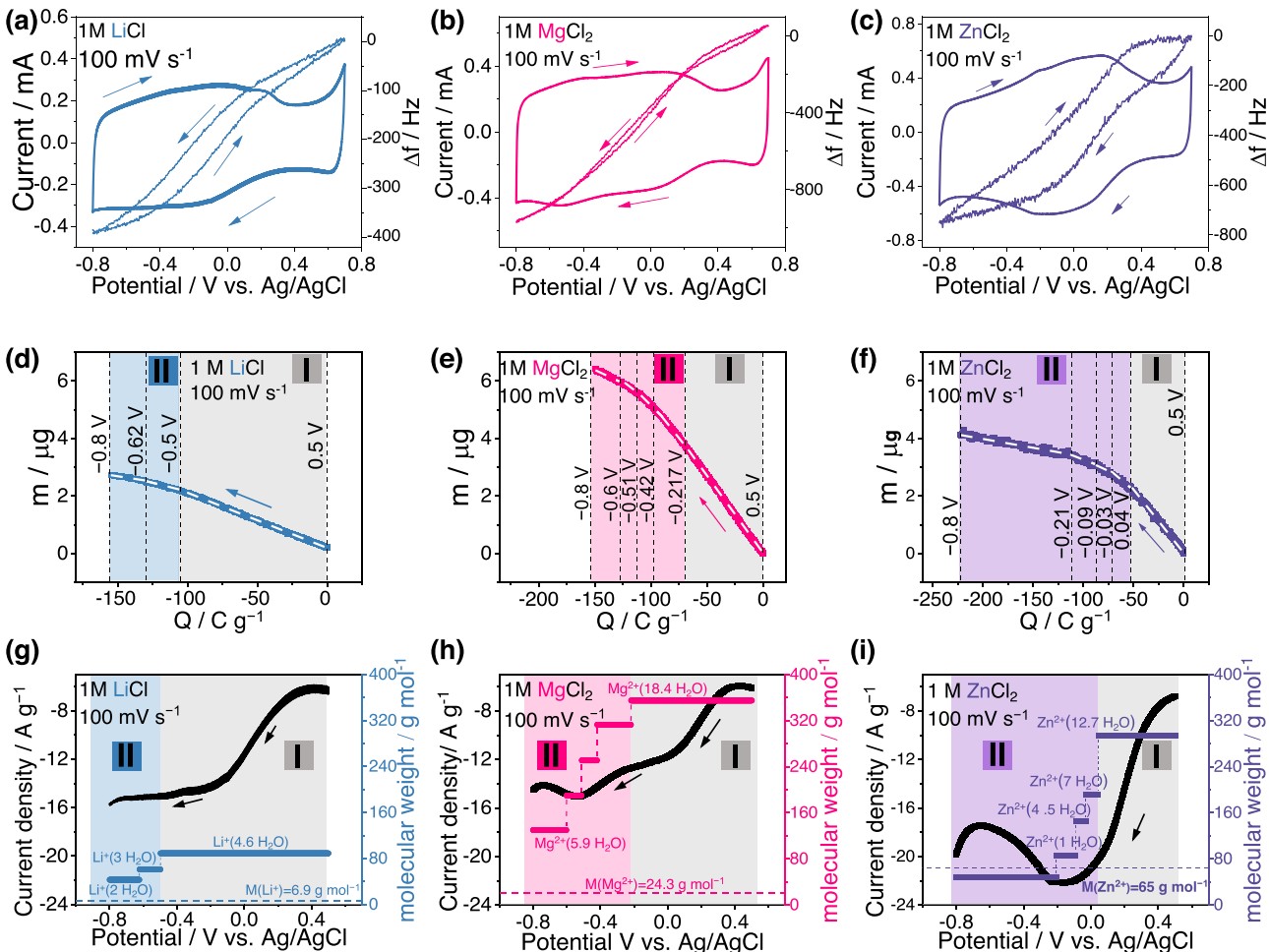

**Fig. 3 | Electrochemical quartz crystal microbalance (EQCM) measurement of rGO in three different electrolytes.** The electrolytes are 1 M LiCl (**a**, **d**, **g**), 1 M MgCl$_2$ (**b**, **e**, **h**) and 1 M ZnCl$_2$ (**c**, **f**, **i**), the potential scan rate is 100 mV s$^{-1}$. **a**, **b**, **c** Cyclic voltammetry curves with corresponding frequency change (Δf), indicated by arrows representing the polarization direction and Δf change direction. **d**, **e**, **f** Electrode mass change (Δm) is plotted against accumulated charges (ΔQ), with two distinct regions shaded in different colors. The white dashed lines show the multi-linear fitting results, with cut-off potentials of each scale separated in vertical black dashed lines. Region I is highlighted in light gray and region II is delineated in blue (Li$^+$), pink (Mg$^{2+}$), and light purple (Zn$^{2+}$). **g**, **h**, **i** Current and equivalent molecular weight obtained from the slope of Δm·ΔQ plots by Faraday's equation is plotted vs. potential, presented in the same two regions as depicted in (**d**–**f**). Dashed lines in (**g**–**i**) represent the theoretical molecular weight of the naked cations.

When moving to Region II, the calculation of the molar weight from the Δm·ΔQ plots highlighted the existence of a dehydration process at larger cathodic potentials for the highly solvated cations (Mg$^{2+}$ and Zn$^{2+}$), while minor dehydration was observed for Li$^+$. This progressive dehydration seems to be associated with the presence of a current broad peak (bump). The hydrated Mg$^{2+}$ discards of a part of its solvation shell, transitioning from Mg$^{2+}$ (18.4 H$_2$O) to Mg$^{2+}$ (5.9 H$_2$O), which well aligns with the current bump situated at −0.5 V (Fig. 3h). Similarly, the Zn$^{2+}$ current peak located at −0.3−0 V corresponds to a substantial removal of its solvation shell (Fig. 3i). Meanwhile, the current for Li$^+$ remained steady in Region II, as a result of minor structural change in the Li$^+$ solvation shell. To summarize, the pronounced dehydration observed for Mg$^{2+}$ and Zn$^{2+}$ during high cathodic polarization (Region II) is assumed to be at the origin of the observed current bumps. This might be due to the reduced effective ion size after removing solvation shells, allowing for a more efficient screening of the electrode charge by the adsorbed ionic species, as desolvated cations are expected to get closer to the carbon surface favoring specific interactions[9,10,51].

The potential-driven step dehydration process provides intriguing insights into the dynamics of cation-electrode interactions and their impact on the electrochemical behavior of the system. The

current bump for Mg$^{2+}$ being located at a more negative potential (−0.5 V) versus that observed for Zn$^{2+}$ (−0.1 V), it suggests a more substantial energy barrier for the electrochemical adsorption of Mg$^{2+}$, resulting in easier Zn$^{2+}$ dehydration and subsequent adsorption into the carbon electrode. This observation is consistent with the zeta potential measurements, suggesting a more robust interaction of Zn$^{2+}$ with rGO. To sum up, the cation-specific electrochemical behaviors studied via both CME and EQCM techniques were found to be closely tied to the interactions between the cations and the rGO electrode. In the case of the Mg$^{2+}$-based electrolyte, the current bump (−0.5 V) arises from the strong Mg$^{2+}$-rGO interaction, often referred as cation-π interaction. This cation-π interaction is even further increased after dehydration, as the discard of thick solvation shell strengthens the Mg$^{2+}$-π bond. This interaction is notably weaker in the Li$^+$-based electrolyte, thus explain the absence of current peaks / bumps for Li$^+$ adsorption in both region I and II. In contrast, Zn$^{2+}$ exhibits a distinct behavior. As a transition metal cation, Zn$^{2+}$ maintains a (+II) oxidation state, mirroring that of Mg$^{2+}$, and sharing comparable hydration free energies and effective ion sizes[25]. Nevertheless, the inherent characteristics of Zn$^{2+}$, especially its filled 3d shell and an available 4 s state, strengthen its interaction with the conjugated π systems of

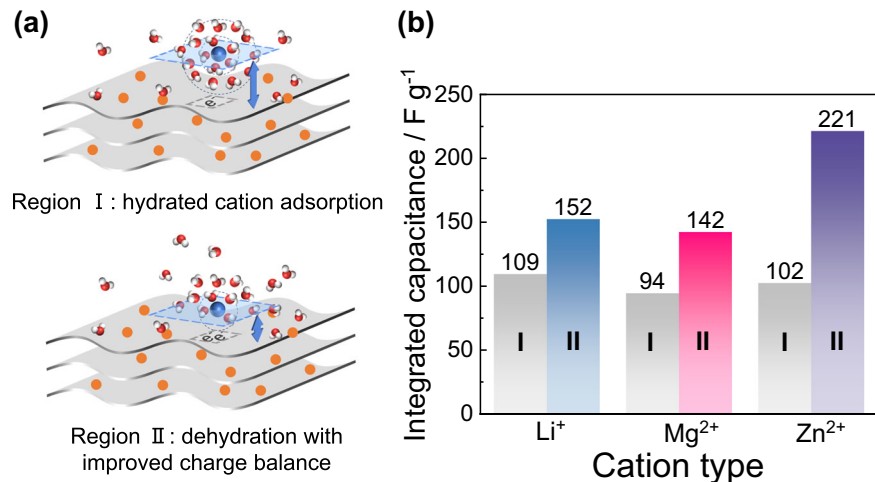

**Fig. 4 | Two regions of charging process and capacitance enhancement. a** Schematic illustration of cation-dependent two regions charge storage process of rGO when applying aqueous near neutral electrolyte. **b** Integrated capacitance of two regions when applying different electrolytes.

rGO[38,39]. This specific interaction is often termed as transition metal-π interaction. Consequently, the stronger $Zn^{2+}$-rGO interaction facilitate $Zn^{2+}$ dehydration and shift the current bump potential to less negative region, causing a confluence of regions I and II, resulting in a broadened current bump in the CV profile.

**Cation desolvation-induced capacitance enhancement**
Figure 4a proposes a schematic illustration of the charge storage processes across the two regions. During cathodic polarization, in region I, solvated or hydrated cations predominantly balance the electronic charges on the rGO surface. This is typical of the electric double layer capacitor (EDLC) behavior, where charge storage happens at the electrode/electrolyte interface without faradaic (redox) processes. Conversely, in region II, cations after losing a part of their solvation shell can approach closer to the electrode surface, leading to more efficient charge screening and storage. This in line with a molecular dynamic study reporting that the more counter-ion interact with the surface, the more the effective counter-charge[10]. These two charging processes were further confirmed by the capacitance analysis (Fig. 4b and Eq. (1)): the nearly constant integrated capacitance ($100 \pm 10$ F g$^{-1}$) calculated in region I for all cations well agrees with the expected EDLC behavior. This region, as suggested, is dominated by the adsorption of solvated cations: the polarizable water molecules of the solvation shell highly screen the ionic charges, making it less effective to balance the electronic charges from electrode side. The dramatic capacitance increase observed in region II can originate from partial desolvation of the cations, that results in a more efficient charge screening thanks to the shortened carbon-cation distance, resulting in higher capacitance.

The difference in capacitance increase observed for the different cations ($Li^+$, $Mg^{2+}$, and $Zn^{2+}$) in region II highlights the specific interactions between each cation and the rGO. The pronounced capacitance enhancement for $Zn^{2+}$ suggests stronger cation interaction with rGO after dehydration. This mirrors the discussions around transition metal-π interactions mentioned previously. The magnitude of enhancement follows the order: $Zn^{2+} > Mg^{2+} > Li^+$, which well aligns with the expected cation-rGO interaction strength. In essence, the findings reiterate the significance of cation hydration and its dynamic interaction with rGO in driving the electrochemical behavior.

**Charge storage active sites and charging process**
Due to the remarkable capacitance achieved in $Zn^{2+}$ electrolyte, we tried to get further understanding about the $Zn^{2+}$ charge storage

mechanism. X-ray diffraction (XRD) measurements were conducted to investigate the active sites. It is worth noting that charge storage sites in reduced graphene oxide (rGO) materials can have two origins: (1) the graphitic-like space situated between the graphene interlayers, corresponding to the $2\theta = 24°$ (001) diffraction peak (Fig. 1b); and (2) the gallery domains formed by the stacking of multilayer rGO particles, characterized by the $2\theta = 11–13°$ diffraction peaks[52]. In our case, rGO particles do not exhibit well defined and organized gallery domains (Supplementary Fig. 6) because of a random stacking of rGO particles. Besides, an operando XRD analysis evidenced the absence of shift of the (002) diffraction peak of rGO during cycling in $Zn^{2+}$-containing electrolyte[42], evidencing that $Zn^{2+}$ cannot intercalate between the graphene interlayers of rGO. Consequently, the charge storage process primarily takes place at the surface of rGO particles in the gallery domains without intercalation.

To get further insights about the charge storage mechanism, we used ECD technique to track the changes in thickness/displacement under polarization. Dilatometry is a well-established technique utilized to measure the expansion or contraction of macroscopic samples. ECD measures the expansion/contraction of a sample during polarization. Figure 5 presents operando ECD plots of the rGO electrode during polarization in different electrolytes. Initially, the pristine electrode, as measured by SEM, is approximately 4 μm-thick (Supplementary Fig. 7). The thin active material coating and the scan rate of 20 mV s$^{-1}$ have been optimized to offset the ECD cell resistance arising from the thick frit separator. Figure 5a–c show the CVs of the rGO samples in the different electrolytes and the electrode displacement (along the z-axis) recorded during the polarization. The displacement was determined by normalizing the thickness change to the pristine thickness of the rGO electrode. Drawing parallels with EQCM data analysis, we examined the cathodic scan from 0.5 V to −0.8 V. In these plots, both accumulated charges and displacement were normalized to zero at 0.5 V. A direct correlation of the (increase of the) displacement with the accumulated charge can be observed, indicating volume expansion due to ion adsorption. Notably, the thickness variation in all three electrolytes were small (below 1 %), which well aligns with a typical EDL-like ion adsorption process[53,54]. Typically, intercalation processes would result in more pronounced volume change[55,56]. A linear volume expansion with charge accumulation was found for both $Li^+$ (Fig. 5d) and $Mg^{2+}$ (Fig. 5e) adsorption. This electrostatic volume expansion results from the transportation of solvated cations from the electrolyte bulk into the capacitive storage sites, which are the spaces formed by the random stacking of rGO particles (Fig. 5g, h). The constant ion sizes result in a continuous linear change of displacement in region I.

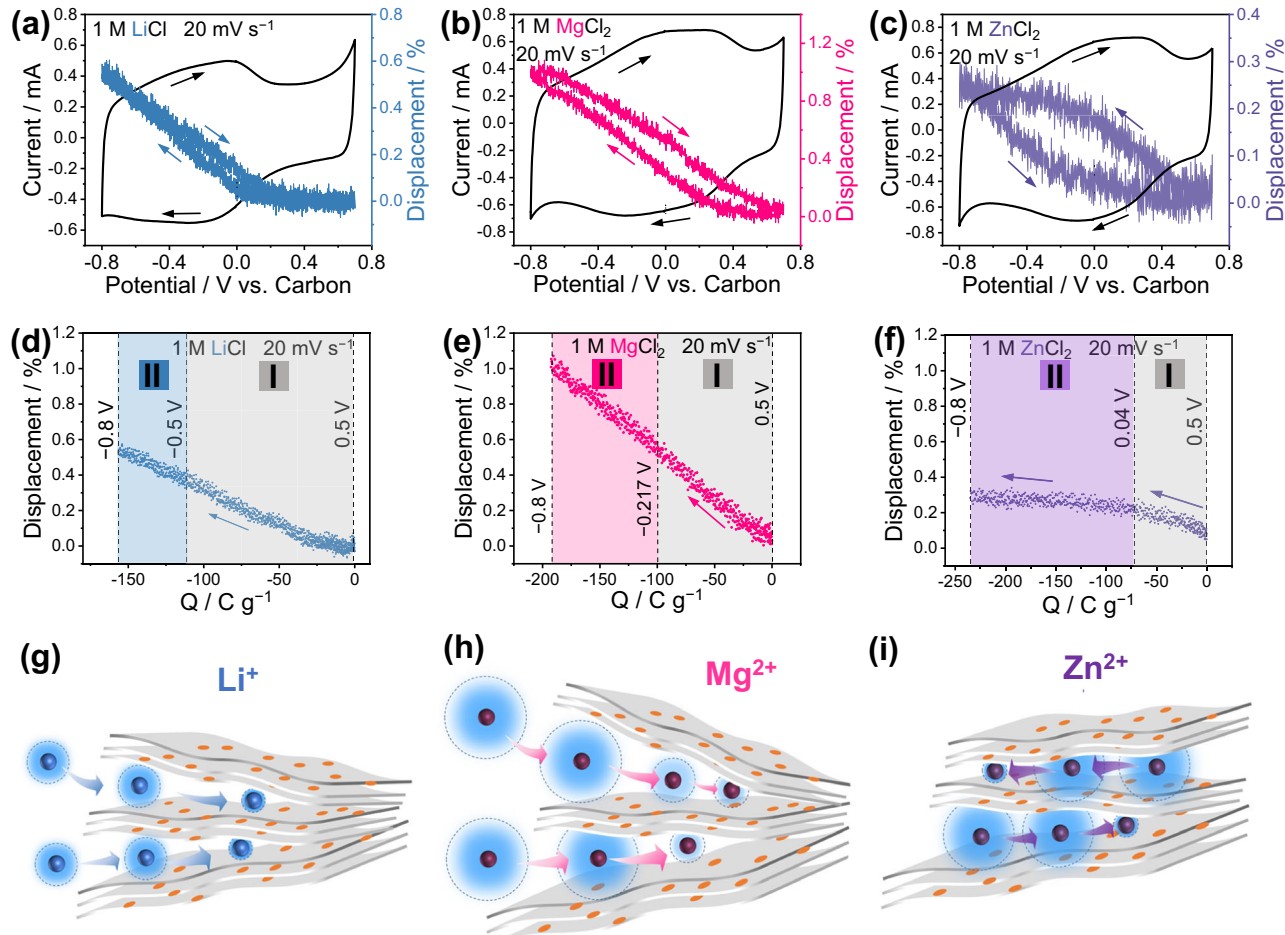

**Fig. 5 | Charge storage active sites and charging mechanism related with cation desolvation.** Dilatometer measurements of rGO with different electrolytes by using a three-electrode operando electrochemical dilatometry cell. **a–c** Cyclic voltammetry curves at scan rate of 20 mV s$^{-1}$ with corresponding displacement change, indicated by arrows representing the polarization direction and displacement change direction. **d–f** Displacement change (also named "height change") of rGO electrodes with different electrolytes at potential scan rate of 20 mV s$^{-1}$.

The zero displacement was set at 0.5 V vs. carbon. **g–i** Schematic illustration depicting the charge storage sites within rGO particles and the charging process accompanied by cation dehydration. Noting that polarization is not depicted in this Figure. The hydration shell encompassing various cations is delineated in varying shades of light blue, showing hydrated cation size changes. Electrolytes are (**a, d, g**) 1 M LiCl, (**b, e, h**)1 MgCl$_2$, and (**c, f, i**)1 M ZnCl$_2$.

Interestingly, the electrode expansion rate remains constant, even when desolvation occurs in region II. This suggest that main effective charges in region II could be the solvated ions already present in the electrode (region I). As a whole, the hydrated cation continues to expand the electrodes at shallow adsorption sites near the edges of multilayer particles in region II and the partial dehydration locally occurs by ion and solvent rearrangement, at less accessible surface, that is inside the particles (Fig. 5g, h)[27,57]. However, the behavior of Zn$^{2+}$ diverges from other cations (Fig. 5f). There is only a minimal volume expansion for Zn$^{2+}$, which may be attributed to the robust binding affinity between Zn$^{2+}$ and rGO. Zn$^{2+}$ are presumably already present in energy storage sites at open circuit potential (Fig. 5i), and the available Zn$^{2+}$ balance the charge more efficiently by partial cation desolvation, accompanied by a minimal expansion. Overall speaking, the charge storage of Zn$^{2+}$ is more efficient than Li$^+$ and Mg$^{2+}$, which also explain the higher average capacitance of Zn$^{2+}$-based electrolyte.

In conclusion, the presented study offers significant insights into the multifaceted electrochemical behaviors depending on cation-rGO interactions, and we found cation-dependent and oxygen-determined specific electrochemical behavior by using CMEs. The charge storage mechanism was firstly studied by EQCM, that suggested the presence of two distinct regions: one dominated by solvated cation adsorption, and the other characterized by cation dehydration, leading to

enhanced charge compensation on the rGO surface. Notably, different cations exhibited varying degrees of interaction strength with rGO, with the sequence being Zn$^{2+}$ > Mg$^{2+}$ > Li$^+$. The dynamic cation-rGO interaction at different polarizations are responsible for the varying electrochemical behavior and energy storage capacity. Furthermore, operando ECD revealed the presence of shallow adsorption sites and accessible, confined sites for Li$^+$ and Mg$^{2+}$ adsorption, resulting in constant volume expansion during charge. Differently, the electrode volume change in Zn$^{2+}$ charge storage mechanism was less, about 0.2 %, suggesting a reorganization of Zn$^{2+}$ present in the electrode at OCV, by solvent reorganization, resulting in partial desolvation. Such findings underscore the potential for tailored electrolyte and electrode design, aiming to optimize capacitive charge storage for high-performance supercapacitors, batteries, and other energy storage systems.

## Methods
### Material preparation
Graphene oxide (GO) was firstly prepared by a modified Hummers method[58]. Then 1.0 mg mL$^{-1}$ of GO aqueous dispersion was used as the precursor to prepare the reduced graphene oxide (rGO) by hydrothermal reduction at 160 °C for 6 h and followed by freeze drying. Afterwards, the further reduced sample (r$^2$GO) was made from rGO

powder by an annealing thermal reduction at 900 °C in argon atmosphere for 1 h, with a heating rate of 10 °C min⁻¹.

## Electrode preparation

First, we commenced the electrode preparation process by thoroughly blending the active powder material, which could be either GO or rGO, together with the conducting additives - carbon black - in a PVDF/NMP solution. The mass ratio used for this combination was 8:1:1 (active material: carbon black: PVDF), and the mixture was stirred overnight. Subsequently, the resulting homogeneous slurry was applied onto the surface of a titanium foil (as current collector) using the doctor blade method. After this step, the electrodes were considered ready for use after vacuum dried at 80 °C for 24 h. Following the drying process, the electrodes were cut into small disks with a diameter of 12 mm. The prepared electrodes were utilized in both three-electrode Swagelok cells and in-situ dilatometry measurements, with a loading mass of approximately 0.7 mg cm⁻².

## Electrochemical tests

The cyclic voltammetry (CV) experiments were conducted using the Biologic VMP-300 electrochemical workstation (France), employing a three-electrode cell setup. The potential range in these CV tests spanned from −0.8 to +0.7 V vs. Ag/AgCl (saturated in a KCl solution). This potential range remained consistent across all our experiments, encompassing CME tests, operando EQCM, and operando ECD measurements.

The choice of scan rate varied based on factors such as the electrochemical cell resistance and the loading mass of the electrodes. For instance, in the CME configuration, where the ohmic drop was negligible, we adopted a high scan rate of 500 mV s⁻¹. This scan rate aimed to enhance current amplification that comes from the active materials and minimize background current. Similarly, in the case of operando EQCM measurements, a slightly reduced scan rate of 100 mV s⁻¹ was employed due to the higher loading mass (in the microgram scale) when compared to the CME test, which featured a loading mass in the nanogram range. For the operando ECD cell, the higher cell resistivity was primarily attributed to the thick frit serving as a separator, making it the most resistive among the different cell configurations, including Swagelok cells, CME setup and EQCM cells. Consequently, a relatively lower scan rate of 20 mV s⁻¹ was selected. It's worth emphasizing that the choice and careful adjustment of these scan rates, tailored to the specific characteristics of each cell configuration, played a crucial role in elucidating the cation-dependent electrochemical behaviors.

The reference electrode for operando ECD tests is the YP-50F activated carbon electrode film, which has similar potential ($\Delta U = 50$ mV) with Ag/AgCl electrode that we used. And the potential of YP50F reference electrode is relatively stable and not polarizable[59].

## Capacitance calculation

The capacitance of rGO was calculated in specific regions of interest. In each region, we integrate of current with respect to time, and subsequently divided it by the potential window of each region, as indicated by Eq. (1):

$$C = \frac{\int_0^t i dt}{V m} \tag{1}$$

where $C$ is the gravimetric capacitance (F g⁻¹), $t$ is the recorded time range (s), $i$ is the response current (A), $V$ is the potential window (V), and $m$ is the active material mass (g).

## Electrochemical quartz crystal microbalance (EQCM) measurements

To prepare the EQCM samples, Au-coated quartz crystals (basic oscillating frequency of 9 MHz, AWSensors, Spain) were homogenously coated with a slurry comprising 90 wt.% of active material rGO, 10 wt.% of polyvinylidene fluoride (PVDF, provided by Arkema), and dissolved in N-Methyl-2-pyrrolidone (NMP, Sigma-Aldrich). The slurry concentration is 1 mg mL⁻¹ (taking all solid mass into account). The coating process was conducted using a spry gun, with a distance 10–15 cm away from the quartz surface. 170 μL of the slurry was added into the spry gun for each quartz, and the coated quartz was vacuum dried overnight at 80 °C before use. The loading mass of the active material coated on the quartz surface is around 20–40 μg cm⁻². The calibration plot (frequency vs. mass for different weight loading) as shown in Supplementary Fig. 8, indicate that the rGO weight loading selected for EQCM experiments falls within the linear change of frequency vs. mass, evidencing the presence of a rigid rGO coating in air. Subsequently, the 3-electrode EQCM setup was applied with the coated QCM quartz severed as working electrode, the platinum wire functioned as counter electrode, and Ag/AgCl employed as reference electrode. These three electrodes were placed within a glassware container and immersed in different aqueous electrolyte that we studied.

All EQCM electrochemical measurements were performed using a Maxtek RQCM system in conjunction with the Biologic potentiostat for simultaneous EQCM and electrochemical measurements. The EQCM data was analyzed according to the Sauerbrey equation[45] (Eq. (2)):

$$\Delta m = - C_f \Delta f \tag{2}$$

where $\Delta m$ represents the change in mass of the coating, and $C_f$ denotes the sensitivity factor of the crystal. The sensitivity factor for the coated quartz was determined through a copper deposition experiment conducted in a solution containing 10 mM CuSO₄ mixed with 0.5 M H₂SO₄. In the stable CV cycles, $C_f$ was calculated as 6.98 ng·Hz⁻¹ (or 5.43 ng·Hz⁻¹ cm⁻², considering the Au crystal electrode surface area of 1.28 cm²). For consistency in results, a few cycles were run prior to initiating EQCM measurements to ensure that the data began from stable and reproducible electrochemical signatures.

For calculating the molecular weight from Δm-ΔQ plots, we apply the Faraday's law (Eq. (3)):

$$\frac{\Delta m}{\Delta Q} = \frac{M_W}{nF} \tag{3}$$

where $\Delta m$ represents the change in mass of the coating, and $\Delta Q$ was the accumulated charges obtained by integrating current with time, $n$ is the ion valence number and $F$ is the Faraday constant (96 485 C mol⁻¹), then $M_w$ is the molecular weight. In our case, $\Delta m/\Delta Q$ is the slope of Δm-ΔQ plot, we consider the charge carriers are Li⁺, Mg²⁺ and Zn²⁺ in Fig. 3, so the $n$ is 1 for Li⁺ and 2 for Mg²⁺ and Zn²⁺ to calculate the molecular weight.

## Other characterizations

XRD data were collected a D4 X-ray diffractometer (Bruker, Germany) equipped with CuKα radiation (λ = 0.154 nm). Scanning electron microscope (SEM) measurements were conducted on TESCAN VEGA3. Zeta potential was determined by Zetasizer Nano ZS90 with (Malvern Co., UK). Samples were prepared by preparing rGO dispersions in 0.01 M salt solutions containing LiCl, MgCl₂ and ZnCl₂. All the samples present similar pH at around 6 as tested by pH paper. Zeta potential was carried out every 10 measurements for each sample and every 20 times scan for each measurement. The mean value and distribution as indicated by the phase plot are good, then we present the mean value among 10 times measurements with error bars. The in-situ displacement of electrode was measured by an ECD-3-nano electrochemical dilatometer, the dilatometer was placed inside an oven with fixed temperature at 25 °C during the whole measurement. TPD-MS measurement was conducted under the Ar atmosphere 75 mL min⁻¹. rGO

powder was placed in a thermo-balance and heat up to 900 °C at a heating rate of 10 °C min⁻¹. The resulted decomposition products were monitored by online mass spectrometry (Skimmer, Netzsch, Germany).

### Reporting summary

Further information on research design is available in the Nature Portfolio Reporting Summary linked to this article.

## Data availability

The data that support the findings of this study are available from the corresponding author upon reasonable request.

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

## Acknowledgements

K.G. was supported by a grant from the China Scholarship Council. H.S. thanks the National Natural Science Foundation of China (Grant no. 22309202). P.S. and P.-L.T are grateful for support from the European Research Council (ERC) and Réseau sur le Stockage Electrochimique de l'Energie (RS$_2$E) and the LABEX STOREX. This research was funded by ERC Synergy Grant MoMa-Stor #951513. For the purpose of open access, the author has applied a Creative Commons Attribution (CC BY) licence to any Author Accepted Manuscript version arising.

## Author contributions

K.G., H.S., P.-L.T., and P.S. designed the research, P.-L.T., and P.S. supervised the project. K.G., and H.S. performed the material synthesis, electrochemical test, data analysis, and composed the manuscript. E.R.-P. carried out the TPD-MS measurement and performed data analysis. All authors discussed and interpreted the results and contributed to the writing of the manuscript.

## Competing interests

The authors declare no competing interests.
