## [Peer Review File · Nature Communications]

Cation Desolvation-induced Capacitance Enhancement in Reduced Graphene Oxide (rGO)REVIEWER COMMENTS

Reviewer #1 (Remarks to the Author):

This paper discusses charge storage mechanisms of reduced graphene oxide in aqueous solutions and the role of ion desolvation for energy storage applications. The identification of the predominant charge storage mechanism in energy storage materials and devices is of utmost importance for the correct characterization of the system. Therefore, the conducted study of molecular-level processes at the electrochemical interface are highly relevant and might lead to better design rules for modern batteries and capacitors, involving a mix of charge storage mechanisms.

I suggest the following moderate revisions and clarifications before publication:

1) Introduction: The difference between capacitive and pseudocapacitive charge storage is not described clearly. Since pseudocapacitance is an often misunderstood term, it would be beneficial if the authors could clarify that pseudocapacitance is faradaic by nature and involves the transfer of charges across the electrochemical interface as opposed to true capacitive charge storage.

The difference of capacitive and pseudocapacitive charge storage has been described by:

S. Fleischmann, J.B. Mitchell, R. Wang, C. Zhan, D. Jiang, V. Presser, V. Augustyn, Pseudocapacitance: from fundamental understanding to high power energy storage materials, *Chem. Rev.* 120 (14) (2020) 6738–6782.

T. Schoetz, L.W. Gordon, S. Ivanov, A. Bund, D. Mandler, R.J. Messinger, Disentangling faradaic, pseudocapacitive, and capacitive charge storage: A tutorial for the characterization of batteries, supercapacitors, and hybrid systems, *Electrochim. Acta* 412 (2022) 140072.

2) Cyclic Voltammetry: Was the charge storage mechanism (faradaic vs. capacitive) determined solely by the shape of the CV? In this case, pseudocapacitive and true capacitive charge storage might not be distinguishable. A more quantitative and rigorous distinction can be made by variable-rate CV analysis as described by Wang et al. (*J. Phys. Chem. C* 2007, 111, 40, 14925–14931).

In that context, the determined capacitance at rGO has a time dependence and is thus not the same as the capacitance C in a true capacitor which would be a constant. The authors might want to add this clarification.

3) EQCM: (a) How large is the damping of the quartz crystal with graphite coating and how thick is the graphite layer compared to the electrodes used in the 3-electrode cells? Graphite tortuosity and particle size might differ between the two techniques and influence the charge storage mechanism.

(b) The Sauerbrey relation can only be applied to flat and rigid surfaces. A graphite surface cannot be considered as rigid and flat. However, the assumptions can be simplified to this model if a fudge-factor is included. If it can be shown that all tested graphite quartz crystals have the same roughness and density, the factor can be seen as a constant and a qualitative comparison can be made.

(c) It would be beneficial for the readership to include a more detailed coating procedure of the quartz crystal if it maintains a reasonable damping value, e.g. coating time, number of layers/loading, distance spray gun to crystal surface and so on.

Reviewer #2 (Remarks to the Author):

This work presents a holistic study of the charge storage mechanism of the rGO model material in near neutral electrolytes. Multiple characterization approaches, such as CME, EQCM and ECD, have been applied to evaluate the electrochemical performance and the mass and volume dynamic changes. It is found that there are two regions with different mechanisms: one dominated by cation adsorption and the other by cation desolvation, dependent on the cation-carbon interaction. These insightful results will generate broad interests to the research community of energy storage. Meanwhile, the manuscript is well-written and the data is nicely interpreted. Therefore, I recommend its publication after the following points are addressed.

1. In Figure 2, a discrepancy in current magnitude is seen for rGO and r2GO using the CME approach.

Given the amount of (oxygen-containing) functional groups and highly packed format, would the wettability be a critical factor leading to the different electrochemical behaviors for these two materials (especially the scale of capacitive background)?

2. It is a bit difficult to follow Supplementary Figure 1 (the axes in the middle) and the authors are suggested to improve it.

3. In Supplementary Figure 3, why does the bumps disappear in (b), since they are still seen under 100 (Figure 2) and 500 (Supplementary Figure 2) mV s^{-1} ? Would the scan rate affect the bumps? Also, have the authors investigated the effect of cation concentration on the electrochemical behaviors?

4. As the authors explained in the main text, it makes sense that a high energy barrier is expected for Mg^{2+} adsorption than Zn^{2+} , which leads to a current bump at more negative potentials. However, by experiment, this is not always the case. As in Figure 5, almost identical CVs are seen for Mg^{2+} and Zn^{2+} . Could the authors elaborate on the possible shift of current bumps constantly observed for the same cation yet among different CV measurements?

5. The authors mentioned 'pseudocapacitive or intercalation processes would result in more pronounced volume change,' while attributing the small volume expansion to ion adsorption. Could they clarify what the difference between ion adsorption in interlayers (as schematically shown in Figure 5g-i) and intercalation is?

6. The redox peaks observed in the range of -0.2 V to 0.2 V on the CV recorded in LiCl tend to be a reflection of the pseudocapacitive behavior of surface functionalities. This can also introduce a volume expansion (as seen in Figure 5a), and then how to distinguish this contribution from that of ion adsorption to the total volume change?

7. Some minor formatting issues need to be considered:

(a) the reference titles would look better if the first letters of the words are capitalized, in a consistent way.

(b) 'Angew. Chem. Int. Ed.' should be the correct form of abbreviation, instead of 'Angew. Chem. Int. Edit.'

(c) 'it's worth mentioning' is better changed to 'it is worth mentioning'. The same applies to 'There is only a minimal volume...'

(d) a full stop should be added after 'by integrating the CV with Eq. (1)'.

(e) 'see supporting information Eq. (2)' should be changed, since it is affiliated at the end of the main text, not in the supporting information. And the other half bracket is missing.

(f) 'the discard of thick solvation shell strengthens the...'

Answer to Reviewers

We really appreciate the valuable comments from the reviewers, which were useful to improve the quality of our manuscript. Our responses to questions (bold) are given in black color and the changes/additions to the manuscript are highlighted in yellow in the main text.

Reviewer #1

Comments: This paper discusses charge storage mechanisms of reduced graphene oxide in aqueous solutions and the role of ion desolvation for energy storage applications. The identification of the predominant charge storage mechanism in energy storage materials and devices is of utmost importance for the correct characterization of the system. Therefore, the conducted study of molecular-level processes at the electrochemical interface are highly relevant and might lead to better design rules for modern batteries and capacitors, involving a mix of charge storage mechanisms. I suggest the following moderate revisions and clarifications before publication:

We are very grateful to the thoughtful and positive comments and appreciate the recognition of the importance of this work.

1. Introduction: The difference between capacitive and pseudocapacitive charge storage is not described clearly. Since pseudocapacitance is an often misunderstood term, it would be beneficial if the authors could clarify that pseudocapacitance is faradaic by nature and involves the transfer of charges across the electrochemical interface as opposed to true capacitive charge storage. The difference of capacitive and pseudocapacitive charge storage has been described by: S. Fleischmann, J.B. Mitchell, R. Wang, C. Zhan, D. Jiang, V. Presser, V. Augustyn, Pseudocapacitance: from fundamental understanding to high power energy storage materials, *Chem. Rev.* 120 (14) (2020) 6738–6782. T. Schoetz, L.W. Gordon, S. Ivanov, A. Bund, D. Mandler, R.J. Messinger, Disentangling faradaic, pseudocapacitive, and capacitive charge storage: A tutorial for the characterization of batteries, supercapacitors, and hybrid systems, *Electrochim. Acta* 412 (2022) 140072.

We agree with the comment and, the difference between capacitive and pseudocapacitive charge storage has been highlighted in the Introduction part, together with the suggested references.

Changes to the manuscript:

“Classical EDL-type charge storage mechanism is achieved by ion adsorption/desorption through electrostatic forces without involving any redox charge transfer. This characteristic ensures fast charging rate, no diffusion limitation and long-term cycling stability.² Pseudocapacitive electrodes exhibit similar electrochemical response to EDL-type electrodes like porous carbon and graphene, with a charge changing linearly with the potentials.³⁻⁴ However, pseudocapacitive charge storage involves charge transfer or partial charge transfer across the electrochemical interface, distinguishing such process from the electrostatic capacitive charge storage observed in EDL-type electrodes,⁴ which in turns leads to a lower cycling stability”

“These recent studies suggests that capacitive charge storage under confined geometries cannot be simply described as either pure EDL or pseudocapacitance, but instead should be considered more as a continuum based on the solvent-mediated interactions between electrolyte ions and electrode host materials.^{13, 17”}

“To understand the charge storage at confined nanoscale interface, a systematic exploration and deeper comprehension are highly desired.”

2. Cyclic Voltammetry: Was the charge storage mechanism (faradaic vs. capacitive) determined solely by the shape of the CV? In this case, pseudocapacitive and true capacitive charge storage might not be distinguishable. A more quantitative and rigorous distinction can be made by variable-rate CV analysis as described by Wang et al. (J. Phys. Chem. C 2007, 111, 40, 14925–14931). In that context, the determined capacitance at rGO has a time dependence and is thus not the same as the capacitance C in a true capacitor which would be a constant. The authors might want to add this clarification.

We thank the Referee for the important comment regarding capacitive and pseudocapacitive contributions.

As mentioned by the Referee, it is very difficult to distinguish between capacitive and pseudocapacitive behaviors. In their seminal paper, B. Dunn’s group have presented a method to distinguish between diffusion limited reactions (bulk redox process) and non-diffusion limited reactions (surface processes) by using the equation $i = k_1 + k_2 v^{0.5}$ instead of $i = a v^b$, where a and b are fitting parameters. The b -exponent is expected to land between 0.5 and 1. If the current is limited by the semi-infinite diffusion of the reactive species, $b = 0.5$, for example battery-type electrodes. Differently, b reaches 1 when the current is surface-controlled, i.e. surface process. Dunn’s method only makes sense if b value is not close to 1, which is not the case (see figures below), even at different potential. This is why capacitive and pseudocapacitive contributions cannot be easily distinguished even using B. Dunn’s approach.

However, the set of results reported here makes the link between these two processes. We show that the current bumps are associated with a partial dehydration of cations during adsorption, that can get closer to the carbon surface and increase the capacitance. This suggests that a continuous transition from double layer to pseudocapacitance can be achieved by dehydration (desolvation), resulting in increased capacitance and current bumps. This also perfectly matches with the concept of electrochemistry under confinement (Fleischmann et al., *Nature Energy* 2022, 7(3): 222-228.).

Following the suggestion from the Referee, we plotted the b -values for different potentials showing it close to 1, among various electrolytes, suggesting current linear increase with scan rate and confirming that the charge storage is driven by a surface charge storage process without diffusion limitation.

Supplementary Figure 3. Cyclic voltammetry curves at different scan rates (0.1, 0.2, 0.5, 1.0, and 2.0 V s⁻¹) of rGO (a-c) and the b -value obtained by linear fitting of the logarithm of the cathodic bump/peak current against the scan rate at different potentials (d-e) when applying different electrolytes in three-electrode cavity micro-electrode configuration. The electrolytes are (a) (d) 1 M LiCl; (b) (e) 1 M MgCl₂; and (c) (f) 1 M ZnCl₂.

Changes to the manuscript:

Supplementary Figure 3 and the associated discussion has been added to the SI part. The other Supplementary Figures have been renumbered.

3. EQCM:

(a) How large is the damping of the quartz crystal with graphite coating and how thick is the graphite layer compared to the electrodes used in the 3-electrode cells? Graphite tortuosity and particle size might differ between the two techniques and

influence the charge storage mechanism.

Thank you for your question. Firstly, we want to clarify that the active material that we investigated in the whole manuscript is the reduced graphene oxide rGO (see the broad 002 XRD peak in Figure 1b).

As the Referee mentioned, the frequency damping is a key parameter to control either or not a gravimetric approach applies (Sauerbrey's equation), when using EQCM. Our EQCM-A (admittance) allows measuring on-fly the motional resistance (R_m) corresponding to the resonance frequency associated with the Butterworth-Van Dyke (BVD) model (*Review of Scientific Instruments* 2002, 73(7): 2724-2737.). In the present study, the motional resistance, R_m , of the quartz crystal after coating rGO is measured $\sim 100 \Omega$ in air, suggesting the coating can be assumed to be rigid. The calibration plot (frequency vs. mass for different weight loading) has been added as Supplementary Figure 8, showing that the rGO weight loading selected for EQCM experiments falls within the linear change of frequency vs. mass, evidencing the presence of a rigid rGO coating in air. After immersion in various electrolytes, R_m for rGO-coated quartz slightly increases (in the 200 - 500 Ω range, see Supplementary Figure 5) but i) it is in the same range of that of bare quartz ($R_m^{\text{bare}} = 150\text{--}300 \Omega$) under the same operating conditions, and ii) as R_m is not in the range of $k\Omega$, so that the dissipation is minimized. The Sauerbrey model then applies to our measurements.

Supplementary Figure 8. Calibration plot of the coating in air (frequency vs. mass for different loadings). The rGO weight loading selected for EQCM experiments falls within the linear change of frequency vs. mass, evidencing the presence of a rigid rGO coating in air.

More details regarding the validity of the gravimetric approach used here are given in the answer to the next question (b) below, as it also deals with the same question.

The thickness of electrodes used in 3-electrode Swagelok cells is around 4 μm , measured by SEM (Supplementary Figure 7), the loading mass is $\sim 0.7 \text{ mg cm}^{-2}$.

Differently, a very thin layer of rGO has been coated on the quartz surface, with a loading mass of $\sim 20\text{--}40 \mu\text{g cm}^{-2}$, which is mandatory to stay within the validity of the Sauerbrey's equation (rigid coating, see above). Assuming a similar packing density of rGO electrode in Swagelok cell and EQCM cell, the average thickness of rGO coating on quartz surface is estimated to be 100–200 nm. The tortuosity effect for such thin coatings is not expected to play an important role.

However, the Referee raised an important point that is not often highlighted in papers. Doing EQCM for more than 10 years, we are fully aware about the importance of the electrode or coating preparation to get reliable and reproducible data. This is why the same batch of rGO material was used in the different experiments (EQCM and Swagelok cells), with same particle size (or particle size distribution) and morphology. Similar electrode preparation procedure for the EQCM (coating by spray gun) and 3-electrode Swagelok cell electrodes (coating by doctor blade), so that the tortuosity and particle size of the rGO electrodes should be similar. We are thus very confident that both rGO electrodes are very similar in both testing techniques.

Changes to the manuscript:

The calibration plot (frequency vs. mass for different weight loading) has been added as Supplementary Figure 8 with the following comment.

“Supplementary Figure 8 shows that the rGO weight loading selected for EQCM experiments falls within the linear change of frequency vs. mass, evidencing the presence of a rigid rGO coating in air.”

(b) The Sauerbrey relation can only be applied to flat and rigid surfaces. A graphite surface cannot be considered as rigid and flat. However, the assumptions can be simplified to this model if a fudge-factor is included. If it can be shown that all tested graphite quartz crystals have the same roughness and density, the factor can be seen as a constant and a qualitative comparison can be made.

Indeed, Sauerbrey's equation was established based on flat and rigid surfaces. The motional resistance change, $\Delta R = R_m - R_m^0$, can provide important information about a process since soft films and viscous liquids will increase motional losses and so increase the value of ΔR . (*Langmuir*, 2015, 31(46): 12664-12673.) In the gravimetric model obeying Sauerbrey relation, the motional resistance change (ΔR) should be much smaller than Δf (*Anal. Chem.* 2011, 83, 23, 8838–8848. *Analytical Chemistry*, 2020, 92(20): 13803-13812. *Energy Storage Materials*, 2019, 21: 399-413.). In our case, the motional resistance change (ΔR) is measured as 4.5 Ω , 4 Ω , and 22 Ω ; and frequency change (Δf) is 400 Hz, 1022 Hz, and 740 Hz for 1 M LiCl, 1 M MgCl₂, and 1 M ZnCl₂ electrolytes, respectively (Supplementary Figure 5). The small change of ΔR versus Δf (see Figure below) evidences the presence of a rigid coating and negligible dissipation during electrochemical cycling process, and the relatively large $\Delta f/\Delta R$ value indicates

all the frequency change resulted from the mass change, which well validates the gravimetric model of Sauerbrey's equation of EQCM measurements. (*Electrochimica Acta*, 2000, 45, 3907-3916; *Analytical Chemistry*, 2020, 92(20): 13803-13812; *Biosensors and Bioelectronics*, 2005, 21, 840-848.)

Supplementary Figure 5. Diagram of the resonant motional resistance change (ΔR) and resonant frequency change ($-\Delta f$) for rGO during the cathodic charging with 1M LiCl, 1M MgCl₂, and 1M ZnCl₂ electrolyte. Green line 1 and 2 (reproduced from Ref. S1)^{S1} represent an elastic mass effect and a pure viscosity–density effect, respectively. The negligible ΔR indicate that the motional resistance change during the CVs stands for a rigid behavior of the coating, thus validating the gravimetric analysis of the electrochemical behavior of the rGO coatings in the different electrolytes.

Changes to the manuscript:

Supplementary Figure 5 has been updated with the ΔR vs. $-\Delta f$ plots above, with the following comment: “The negligible change of the motional resistance ΔR vs. $-\Delta f$ evidences a rigid behavior of the coating during polarization, thus validating the gravimetric analysis of the electrochemical behavior of the rGO coatings in the different electrolytes.”

(c) It would be beneficial for the readership to include a more detailed coating procedure of the quartz crystal if it maintains a reasonable damping value, e.g. coating time, number of layers/loading, distance spray gun to crystal surface and so on.

Thanks for the suggestion; we have added more details in the coating process: The slurry concentration is 1 mg mL⁻¹ (taking all solid mass into account). The coating process was conducted using a spray gun, with a distance 10-15 cm away from the quartz surface. 170 μL of the slurry was added into the spray gun for each quartz, and the coated quartz was vacuum dried overnight at 80 °C before use. The loading mass of the active material coated on the quartz surface is between 20–40 $\mu\text{g cm}^{-2}$ in the different

electrolytes.

Changes to the manuscript:

The following text has been added into the Experimental part:

“The slurry concentration is 1 mg mL^{-1} (taking all solid mass into account). The coating process was conducted using a spray gun, with a distance 10-15 cm away from the quartz surface. $170 \text{ }\mu\text{L}$ of the slurry was added into the spray gun for each quartz, and the coated quartz was vacuum dried overnight at $80 \text{ }^\circ\text{C}$ before use. The loading mass of the active material coated on the quartz surface is around $20\text{-}40 \text{ }\mu\text{g cm}^{-2}$.

Reviewer #2

Comments: This work presents a holistic study of the charge storage mechanism of the rGO model material in near neutral electrolytes. Multiple characterization approaches, such as CME, EQCM and ECD, have been applied to evaluate the electrochemical performance and the mass and volume dynamic changes. It is found that there are two regions with different mechanisms: one dominated by cation adsorption and the other by cation desolvation, dependent on the cation-carbon interaction. These insightful results will generate broad interests to the research community of energy storage. Meanwhile, the manuscript is well-written and the data is nicely interpreted. Therefore, I recommend its publication after the following points are addressed.

We would like to thank the Referee for their thoughtful feedback. We appreciate the positive remarks.

1. In Figure 2, a discrepancy in current magnitude is seen for rGO and r²GO using the CME approach. Given the amount of (oxygen-containing) functional groups and highly packed format, would the wettability be a critical factor leading to the different electrochemical behaviors for these two materials (especially the scale of capacitive background)?

The oxygen-containing functional groups play a major role in the electrochemical behavior as reduced rGO (denoted as r²GO) with fewer oxygen surface groups shows limited activity (see Figure 2 and Supplementary Figure 3). This strongly suggests, as the Referee mentions, that the surface wettability is a key factor, allowing access of ions/solvent to active sites and improving the ion transportation as well. Surface wettability is the prerequisite condition to enable a possible electrochemical activity. Then, ion-carbon interaction, desolvation and confinement is the key to achieve electrochemical performance, as shown in our paper.

2. It is a bit difficult to follow Supplementary Figure 1 (the axes in the middle) and the authors are suggested to improve it.

Thanks for pointing the mix in Supplementary Figure 1. We addressed the concern by creating a revised version of the Figure that is clearer and more reader-friendly. The updated figure has been moved into our revised manuscript, ensuring improved visual clarity for the readers.

Change to the manuscript:

Supplementary Figure 1 has been revised and the axis modified.

3. In Supplementary Figure 3, why does the bumps disappear in (b), since they are still seen under 100 (Figure 2) and 500 (Supplementary Figure 2) mV s^{-1} ? Would the scan rate affect the bumps? Also, have the authors investigated the effect of cation concentration on the electrochemical behaviors?

We thank the Referee and apologize for lack of clarity.

The difference between Supplementary Figure 2 and (former) Supplementary Figure 3 (the new Supplementary Figure 4) comes from the different set-up used: a cavity micro-electrode (CME) as working electrode in Supplementary Figure 2 and Swagelok cell in Supplementary Figure 4 (former Supplementary Figure 3 mentioned by the Referee). Briefly, CME allows for using small amount of active powder material (10^{-7} – 10^{-8} g), without the need for conductive agents or binders; ohmic drops are then very small due to the low current (few μA) recorded. Such electrode design allows for running CVs at high potential scan rates (up to few V s^{-1}) to study the electrochemical kinetics within a large range of scan rates, without being limited by ohmic drops. The limited ohmic drops is the reason why, even at high potential scan rates of 500 mV s^{-1} , current bumps are still visible (Supplementary Figure 2). Same applies to EQCM measurements, where the active material loading mass of EQCM is around 20 micrograms, which is about 2 orders of magnitude lower than the electrodes used in Swagelok cell and electrochemical dilatometer (ECD) measurements ($\sim 0.7 \text{ mg cm}^{-2}$).

We also investigated the cation concentration on the electrochemical behavior, as shown in Figure R1. The ionic conductivity measured for the electrolyte with different concentrations are 13 mS cm^{-1} , 1.7 mS cm^{-1} , and 0.9 mS cm^{-1} for 0.1 M, 0.01 M, and 0.005 M Li_2SO_4 electrolyte, respectively. The distorted CV shape observed when decreasing electrolyte concentration is mainly attributed to ohmic drops associated with limited ion transportation at lower ionic conductivity. Meanwhile, the cation-specific electrochemical signal also shifted in potential, as ohmic drops limits accessibility of confined geometries at low electrolyte concentration.

Figure R1. Cyclic voltammetry curves of Li_2SO_4 at different electrolyte concentrations (0.005 M, 0.01 M, and 0.1 M) in a three-electrode cavity micro-electrode configuration.

4. As the authors explained in the main text, it makes sense that a high energy barrier is expected for Mg^{2+} adsorption than Zn^{2+} , which leads to a current bump at more negative potentials. However, by experiment, this is not always the case. As in Figure 5, almost identical CVs are seen for Mg^{2+} and Zn^{2+} . Could the authors elaborate on the possible shift of current bumps constantly observed for the same cation yet among different CV measurements?

As discussed above, the CME is well suited to study the cation-dependent electrochemical behavior thanks to limited ohmic drops, when compared to Swagelok cells and electrochemical dilatometer (ECD) measurements. The obtained CVs using CME are reproducible and similar among different electrolytes containing the same cation, as shown in Supplementary Figure 2.

For ECD measurements (Figure 5), ~1 cm thick T-shape porous alumina frit is used as separator in the cell to efficiently transfer the electrode volume change to the sensor (capacitor). This is the reason why CVs were recorded at 20 mV s^{-1} with ECD set-up, to limit the influence of ohmic drops. The presence of the thick alumina separator thus explains the slight potential shift of ECD CVs. One can say we could have decreased the scan rate but the presence of a thermal drift and displacement drift due to sensor stray capacitance makes the measurement much less accurate.

5. The authors mentioned ‘pseudocapacitive or intercalation processes would result in more pronounced volume change,’ while attributing the small volume expansion to ion adsorption. Could they clarify what the difference between ion adsorption in interlayers (as schematically shown in Figure 5g-i) and intercalation is?

This is an important point and we should have better clarified in the text.

Graphite is characterized by high crystallization, evident in a distinct 002 peak at 26.4° in XRD pattern, corresponding to a well-ordered 002 crystal plane (d-spacing along the c axis of 0.335 nm). The rGO synthesized in our study similarly exhibits a graphitic structure with a 002 peak at $2\theta = 24^\circ$ (Figure 1b) corresponding to an increased d-spacing of 0.37 nm. However, this peak appears broader for rGO than that of graphite, indicating a less ordered structure, as expected from the increase d-spacing in rGO (see for instance *Chemistry of Advanced Materials 2019*, 4(3), 17-26). Then, the broad 002 peak in rGO results from the random stacking arrangement of the multilayer rGO particles. This configuration results in the presence of void spaces (termed as gallery domains in the paper) containing abundant surface/edge sites between the randomly stacked rGO particles. These zones are the active sites for ion adsorption, such as presented in Figure 5.

We compare here ion adsorption to ion intercalation process observed in graphite materials, as indicated in Ref. 53 (*Energy Technol. (Weinh)*2022, 10, 2101120). Ref. 55 (*ECS Transactions*, 2015, 69(22): 9) and Ref. 56 (*Electrochimica acta*, 2017, 257: 423-435). In conventional ion intercalation into graphite materials, the d-spacing of the 002 crystal plane expands, resulting in important volume expansion (> 5%, see Ref 53). This intercalation process also involves slow kinetics due to its bulk nature, which is in stark contrast to what we observed, as the charge storage mechanism in rGO is a fast, non-diffusion limited surface process (see new Supplementary Figure 3), achieved with small volume expansion (<1%, see Figure 5) as expected for ion adsorption (Ref 53). This supports our view that the charge storage mechanism in rGO is achieved by ion adsorption on the active sites present in the galleries (between rGO particles), but not by ion intercalation in the d-spacing of the rGO particles. Also, an in-situ XRD analysis (provided by Ref. 42) evidenced the absence of shift of the (002) diffraction peak of rGO during charging/discharging in Zn²⁺-containing electrolyte. Besides, the cation intercalation into graphitic units always need to desolvation and overcome higher energy barriers, thus normally occurs at more negative potentials, which is not true in our case. The minor volume expansion (Figure 5) and fast ion transport kinetics (Supplementary Figure 3) strongly suggest a surface (ion) adsorption process, but not an ion intercalation one.

Changes to the manuscript:

Supplementary Figure 3 has been added, showing that the charge storage mechanism for rGO electrode is a non-diffusion limited, surface controlled process.

The statement “pseudocapacitive or intercalation processes would result in more pronounced volume change” has been modified to “intercalation processes would result in more pronounced volume change”

The words “in the gallery domains” has been added to the text in the sentence “Consequently, the charge storage process primarily takes place at the surface of rGO particles in the gallery domains without intercalation.”

6. The redox peaks observed in the range of -0.2 V to 0.2 V on the CV recorded in LiCl tend to be a reflection of the pseudocapacitive behavior of surface functionalities. This can also introduce a volume expansion (as seen in Figure 5a), and then how to distinguish this contribution from that of ion adsorption to the total volume change?

Pseudocapacitive behavior of surface functionalities and ion adsorption between both surface processes, it might be difficult to distinguish between their contribution. However, this may not be an important point in the present case.

As already mentioned in the answer to comment #2, surface wettability is the prerequisite condition to enable a possible electrochemical activity. Then, ion-carbon

interaction, desolvation and confinement is the key to achieve improved electrochemical performance (see Figure 4), as different cations result in different electrochemical behavior in zone II, while the oxygen functionalities content is kept the same. More specifically, the volume expansion in LiCl is about 1%, while it is only 0.3% in Zn²⁺-containing electrolytes, that is three times less. It shows that the O-containing groups are definitely not the main contributors to the volume change (and to the capacitance) but ion-carbon interaction, in desolvation and confinement.

Instead, the set of results reported here makes the link between these two processes (EDL and pseudocapacitance). We show by EQCM that the current bumps are associated with a partial dehydration of cations during adsorption, that can get closer to the carbon surface and increase the capacitance. Recent studies highlighted that capacitive charge storage under nanoconfinement cannot be described as either pure EDL or pseudocapacitance, but instead should be considered as a continuum, based on the interaction between the electrolyte ions and the electrode host materials (*Nature Materials*, 2021, 20(12): 1689-1694. *Nature Energy*, 2022, 7(3): 222-228.). Our results perfectly match with such continuous transition from double layer to pseudocapacitance by cation partial dehydration (desolvation), resulting in increased capacitance and current bumps.

7. Some minor formatting issues need to be considered:

- (a) the reference titles would look better if the first letters of the words are capitalized, in a consistent way.**
- (b) ‘Angew. Chem. Int. Ed.’ should be the correct form of abbreviation, instead of ‘Angew. Chem. Int. Edit.’.**
- (c) ‘it’s worth mentioning’ is better changed to ‘it is worth mentioning’. The same applies to ‘There is only a minimal volume...’.**
- (d) a full stop should be added after ‘by integrating the CV with Eq. (1)’.**
- (e) ‘see supporting information Eq. (2)’ should be changed, since it is affiliated at the end of the main text, not in the supporting information. And the other half bracket is missing.**
- (f) ‘the discard of thick solvation shell strengthens the...’**

Thanks for pointing out our format issues. All the suggested changes have been implemented in the manuscript. We also revised the whole paper to correct typos and used appropriate and consistent format for references.

Finally, we would like to thank again the Referee for their positive comments.

REVIEWERS' COMMENTS

Reviewer #1 (Remarks to the Author):

The authors have addressed my comments well and I think that the manuscript can be accepted.

Reviewer #2 (Remarks to the Author):

The authors have now well addressed my concerns and sufficiently answered my questions. Therefore, I would like to recommend this manuscript to be published as it is.